# A nanomaterials discovery robot for the Darwinian evolution of shape programmable gold nanoparticles

Daniel Salley [1,2], Graham Keenan[1,2], Jonathan Grizou[1], Abhishek Sharma[1], Sergio Martín[1] & Leroy Cronin [1✉]

The fabrication of nanomaterials from the top-down gives precise structures but it is costly, whereas bottom-up assembly methods are found by trial and error. Nature evolves materials discovery by refining and transmitting the blueprints using DNA mutations autonomously. Genetically inspired optimisation has been used in a range of applications, from catalysis to light emitting materials, but these are not autonomous, and do not use physical mutations. Here we present an autonomously driven materials-evolution robotic platform that can reliably optimise the conditions to produce gold-nanoparticles over many cycles, discovering new synthetic conditions for known nanoparticle shapes using the opto-electronic properties as a driver. Not only can we reliably discover a method, encoded digitally to synthesise these materials, we can seed in materials from preceding generations to engineer more sophisticated architectures. Over three independent cycles of evolution we show our autonomous system can produce spherical nanoparticles, rods, and finally octahedral nanoparticles by using our optimized rods as seeds.

[1] School of Chemistry, University of Glasgow, Glasgow G12 8QQ, UK. [2] These authors contributed equally: Daniel Salley, Graham Keenan.
✉email: Lee.Cronin@glasgow.ac.uk

The study of nanoparticles has increased vastly due to their unique properties, leading to new developments in many different areas such as surface enhanced Raman scattering (SERS)[1,2], microscopy[3], drug delivery agents[4,5], cancer treatment[5,6], carriers for biomolecules[7], etc[8–10]. For this reason, several synthetic protocols have emerged such as electro-chemical[11,12], photochemical[13], template[14,15], Turkevich[16,17], or seed-mediated growth[18,19], to form different shapes of nano-particles e.g. spheres[20], rods[21–23], cubes[24,25], etc. with a host of different properties. Despite the fact that so many synthetic methods have been developed, these have proven difficult to control and produce large amounts of by-products, as well as having problems with reproducibility that have made the synth-esis of gold nanoparticles quite challenging[10]. This means the ability to precisely control the shape of the nanoparticle, and therefore its physical properties, and application, can be chal-lenging for the discovery and for the process of reproducing the protocol. Indeed, the difficulty in the reproduction of known protocols is a major bottleneck preventing the extended devel-opment and use of such materials.

To address these fundamental issues, we hypothesised that the algorithm-driven discovery and digital control of synthesis using a robotic system could revolutionise the design and control of complex faceted nanoparticles. Indeed algorithms have recently been used in self-optimising chemical reactions[26], exploring cat-alysis[27] and also the nucleation of nanocrystals in microfluidic devices[28]. This is because the robotic system could allow the high-fidelity reproduction of the methods used to discover the nano-particles, and this code could be replayed to generate the clusters again minimising errors. In addition, we wanted to use a genetic algorithm approach that not only uses an electronic genome, but also explores the idea that it is possible to evolve physical objects. These objects are not only improved by evolution towards a target, but then could then be used as physical seeds to help direct to new targets. This means the evolutionary trajectory is also imprinted into the physical object, rather than just being weakly associated in an electronic genome. The idea of embodied evo-lution[29] is mostly confined to robotics, but also has been explored with some materials[30].

As a result, we have developed an affordable and simple semi-batch liquid handling platform that is capable of the exploration and optimisation of a chemical space for the synthesis of gold nanomaterials using in-line UV-Vis spectroscopy. Our synthetic method can explore the chemical space for the synthesis of gold nanoparticles (AuNPs) by utilising a genetic algorithm (GA), as well as the physical products produced as templates or 'off-spring' to seed further explorations. This method, also known as hier-archical evolution, see Fig. 1, consists of the preparation of gold seeds starting from raw chemicals and optimising for shape and distribution using a specified spectral target. The system can work towards a single-peak desired target such as gold nanospheres (AuNSs) and can also use the resultant nanoparticles as seeds to synthesise more complex structures, in our case using optimised gold nanorods (spectral target 2, AuNRs) were used as the seeds to synthesise octahedral nanoparticles (Fig. 1b, c).

## Results

**Workflow.** The first stage of our work began with acquiring an expected UV signal for known gold nanospheres from a com-mercially available source (Sigma-Aldrich) in order to establish the spectral target 1 for the automated system. Next, we synthesised gold nanorods following a synthesis described in the literature[10] in order to obtain spectral target 2 for the platform. The platform begins with a set of reagents in order to synthesise spheres aiming for the designated spectroscopic target using the genetic algorithm.

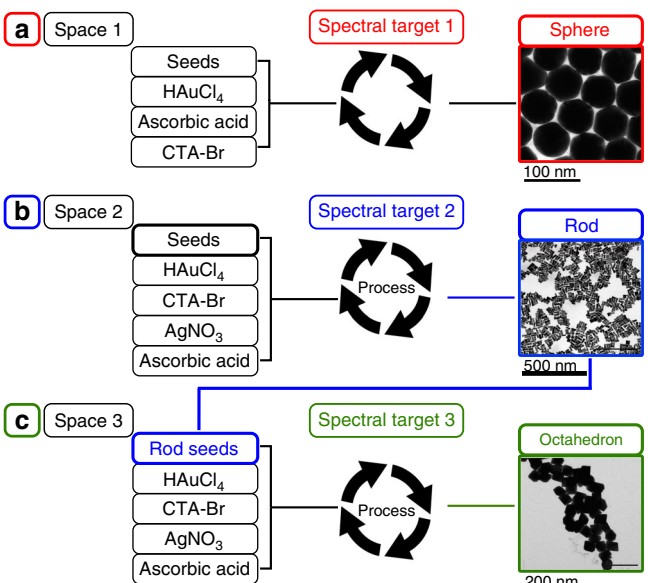

**Fig. 1 Flow diagram of the hierarchical evolution of gold nanomaterials. a** Chemical space 1, containing reagents for the synthesis of spheres, was explored using the platform until spectral target 1 (spheres) was reached/ optimised. **b** Known literature seeds[10] used in chemical space 2 with known reagents for the synthesis of rods until spectral target 2 (rods) was reached/optimised. **c** Target 2 optimised rods used to as seeds to achieve target 3 spectra for unknown nanoparticle shape outcome.

Once the platform has reached the desired target, the optimised particles are analysed. The platform was then given the next set of additional reagents required to synthesise gold nanorods along with the new objective function targeted in the UV-Vis. The ratios of those additional reagents are estimated and optimised by means of the genetic algorithm. Once the system has obtained optimal rods, these particles are used as the seeds in order to synthesise other types of nanomaterials with an objective we set, rather than values based on the literature, see Fig. 2.

**Platform.** The platform is designed to perform a full generation of 15 reactions in parallel and extract samples for UV-Vis analysis once complete, see Fig. 3. At the heart of the robot is a 24° Geneva wheel mechanism which is used to produce 15 movements to complete a full rotation of the reaction vial holder and ensure accurate vial placement under the stationary dispensing stage. The individual reaction vials were stirred directly via custom mounting housings, containing 2 neodymium magnets, which are rotated using a circular array of small 12 V DC fans. The fan speed is determined by pulse width modulation (PWM) con-trolled via Arduino Mega 2560. All mobile and structural com-ponents were 3D printed using an Object500 Connex printer, bought from Openbuilds suppliers, or laser cut from acrylic. The Geneva wheel is powered by a Nema 14 stepper motor and reagents dispensed using Tri-Continent C3000 syringe pumps. Sample extractions to UV-Vis and vial cleaning cycles were achieved via in-house designed modules capable of z-axis movement. All electronic components were controlled by Ardu-ino Mega via in-house software. The platform is housed in an enclosure to control temperature and humidity and the reaction samples were analysed using in-line UV- Vis analysis using an Ocean Optics Flame spectrometer. Further component and build details can be seen in Supplementary Information, Hardware section.

**Algorithm.** In order for the robot to facilitate autonomous synthesis and decision making, an heuristic artificial intelligence

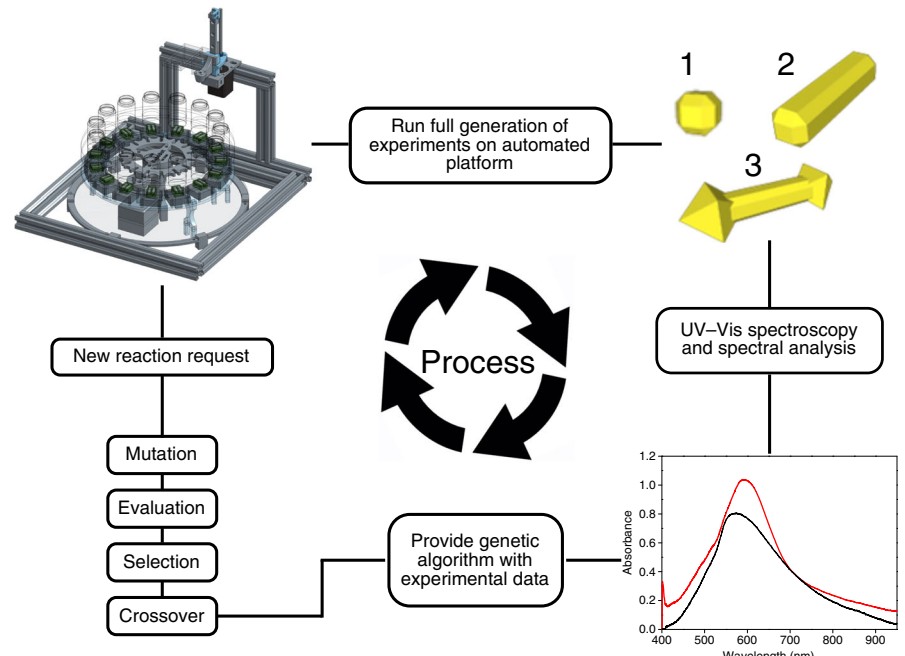

**Fig. 2 Scheme of the platform workflow for the hierarchical evolution of AuNPs.** (proceeding clockwise from top left position) The platform itself (see SI for build details). Each new series of reaction generations aims for a specified spectral target, beginning with a random exploration of the chemical space. Volumes of stock reagents are initially selected at random, dispensed by the platform and analysed by in-line UV-Vis spectroscopy. The resultant spectra are assigned a fitness value and evaluated via our genetic algorithm. The algorithm mutates the experimental parameters and crosses them over attributes of the highest fitness samples to generate new experimental parameters for the next generation. The cycle repeats until the target was reacted, each 15-reaction generation of a given series proceeding towards the predefined spectra.

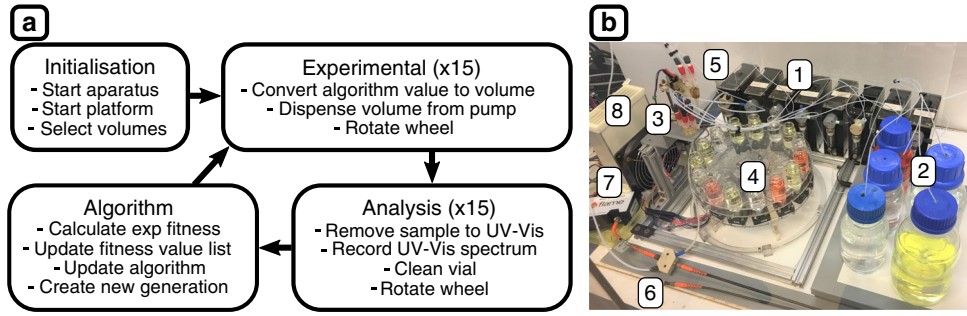

**Fig. 3 General operating outline and top view of the robot for the controlled synthesis of AuNPs. a** Summary of the initialisation of the platform, the experimental and analysis sequences and the algorithm operations for a single generation of reactions. **b** The platform consisting of (1) tri-continent C3000 syringe pumps, (2) reagent bottles on stirring plate, (3) dispensing stage, (4) geneva wheel with vial tray, (5) sample extraction module, (6) flow cell/optics, (7) ocean optics flame UV-Vis spectrometer and (8) heating element.

search method was employed based upon a genetic algorithm (GA)[31]. GAs are often used for finding optimised solutions to search problems inspired, and loosely based, on the theory of natural selection however normally physical material is not passed between optimisation runs. Genetic algorithms are excellent for searching through large and complex data sets and are considered capable of finding reasonable solutions when a large number of variables must be explored.

An initial set of randomly generated parameters are created based on a numerical seed, and the platform executes the experiments and assigns each a fitness value based on the UV-Vis spectra of the samples. These values are then analysed, assessed by the GA, and then a selection process is conducted. This involves selecting which experimental parameters will continue to be used in the next generation. These parameters then undergo a process of recombination – taking parameters from each of these

formulations, splitting and merging them until a new formulation code is generated. During this process, one or more traits of these formulations are randomly modified, creating a "random walk" in the direction of the target solution. These new parameters then replace their counterparts leading to a new set of values for the next generation of reactions, and this continues until the experiment has achieved the predefined spectroscopic target. All the aspects of the platform (hardware control, analysis and optimisation) are controlled via in-house developed software written in Python. The user supplies an experimental configuration file, detailing parameters to use such as experiment type, numerical seed, number of generations to perform, etc.

**Spherical target**. The desired target of space 1 (Fig. 1) was spherical particles, roughly 80 nm in diameter, with a single UV

peak at 553 nm. A seed mediated approach was proposed using known seeds from the literature[10]. The synthesis of these ≈2 nm gold seeds required aqueous solutions of three reagents, $HAuCl_4$, CTAB and $NaBH_4$, and the methods we used followed from the preparation described by Nikoobakht[10]. Briefly, the seeds were synthesised by first mixing CTAB solution (5 mL, 0.2 M), with $HAuCl_4$ (5 mL, 0.0005 M). Under vigorous stirring an ice-cold solution of $NaBH_4$ (0.6 mL, 0.01 M) was then added and the reaction was kept at 30 °C. After 5 min of stirring the seed solution was left for 30 min undisturbed and finally diluted to 30 mL to be used along with aqueous stock solutions of CTAB

(0.2 M), $HAuCl_4$ (0.001 M) and ascorbic acid (0.0058 M) in the reactions of space 1. The algorithm was free to vary all four reagents of the space. In order to calculate the fitness factor for the sphere synthesis, two parameters were considered, that is the absorbance intensity and the distance of the observed peak from the position of the objective, see Fig. 4a. If a sample had absorbance past 650 nm, that was ≥40% intensity of the primary peak, it was given a fitness penalty to promote uniformity in product formation. Details of this fitness calculation can be found in Supplementary information, Software section. Initial experiments for space 1 were performed at 30 °C, stirred for 30 min and

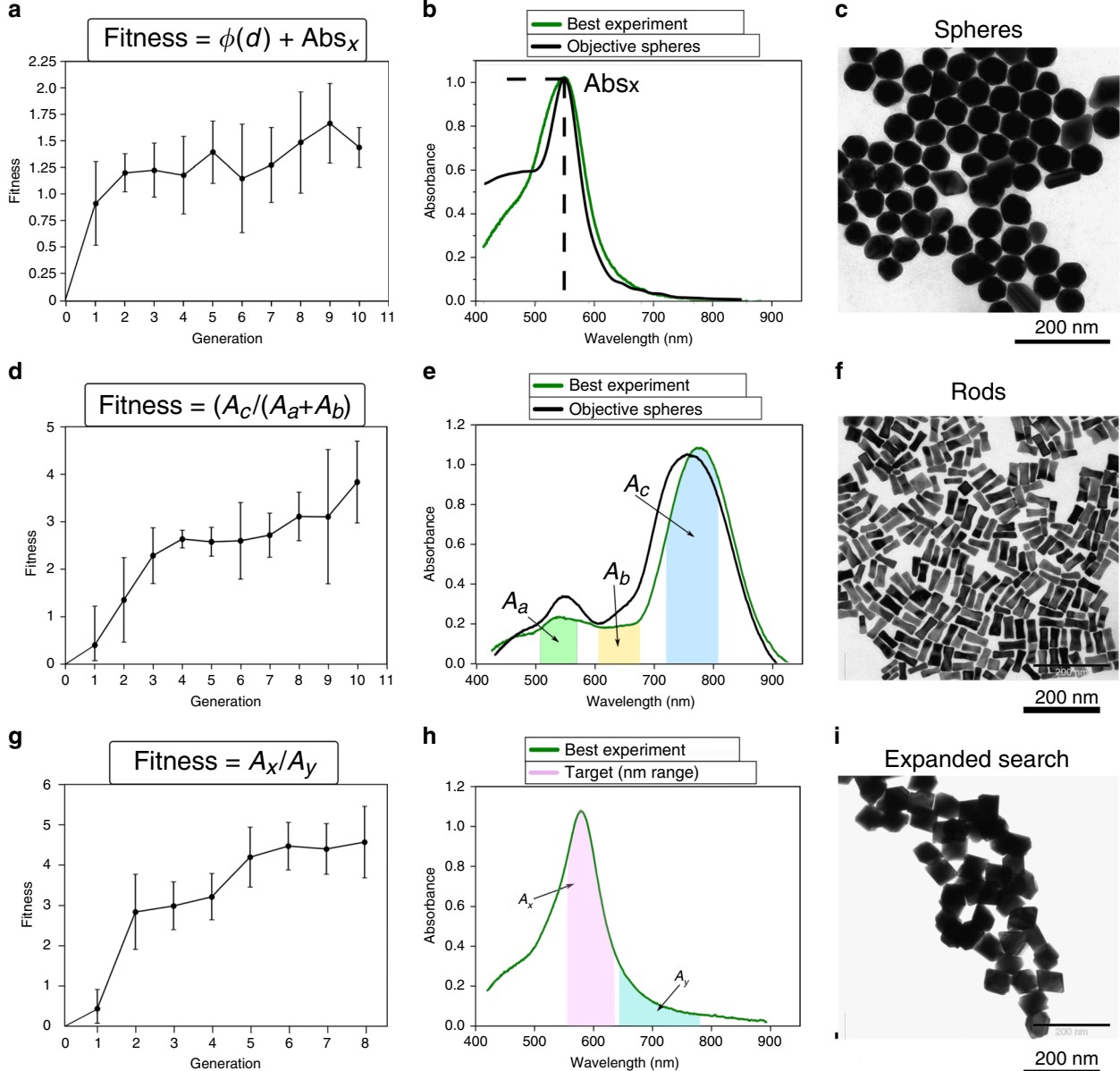

**Fig. 4 Chemical spaces 1–3 with fitness functions applied to several generations resulting in gold nanoparticles evolving from seeds to higher complexity particles. a** Evolution of the median fitness per generation for the nanospheres evolution. **b** Comparison between UV-Vis spectrum of simulated spheres from commercially available sources and the best spheres obtained with the platform. **c** TEM image of the ≈80 nm AuNSs that correspond to the UV-Vis spectrum of the best spheres obtained with the platform. **d** Evolution of the median fitness per generation for the nanorods evolution. **e** Comparison between UV-Vis spectrum of rods described in the literature and the highest fitness AuNR obtained with the platform. **f** TEM image of the AuNRs that correspond to the UV-Vis spectrum of the best nanorods obtained with the platform. **g** Evolution of the median fitness per generation for the expanded search attempting to achieve a single peak at 580 nm. **h** Comparison between UV-Vis target peak wavelength region (pink) set by us and the spectrum with highest similarity obtained with the platform. **i** TEM image of the octahedral shaped gold nanorods that correspond to the highest similarity UV-Vis spectrum obtained with the platform. Error bars represent the standard deviation of the fitness values for a given generation.

left to grow for an additional 60 min before a sample was automatically sent for in-line UV-Vis analysis.

The system then ran for 10 generations, with 15 experiments per generation, and Fig. 4a shows the evolution of the fitness factor towards higher values as a function of generation. This means that the robot, driven by the GA, can progress towards the UV-Vis spectra-based objective through successive generations. The highest scoring sample in terms of fitness can be seen compared to the simulated spectra of commercially available particles of the same size, Fig. 4b. This reaction used the following volumes of the stock solutions listed above: 2.748 mL CTAB, 5.5832 mL $HAuCl_4$, 1.604 mL ascorbic acid and 0.063 mL of seeds. Samples were centrifuged at 10,000 RPM for 10 min and analysis of the precipitated materials using TEM, see Fig. 4c, confirmed the presence of ≈ 80 nm AuNSs as expected.

**Rod target**. Proceeding from spheres to rods as the platform target, the same ≈2 nm seeds as before were used with the addition of $AgNO_3$ as a symmetry breaking agent. To obtain our target spectra the protocol for seed mediated nanorod synthesis described in the literature[10] was performed on the bench. The procedure is as follows; an aqueous solution of CTAB (5 mL, 0.2 M) is added to a solution of $AgNO_3$ (0.15 mL, 0.004 M), followed by $HAuCl_4$ (5 mL, 0.001 M). After gentle mixing, ascorbic acid (70 μL, 0.0788 M) was added and the solution became colourless, and this was the followed by the addition of 12 μL of gold seeds. The solution was kept under constant stirring at 30 °C (±1 °C) and the gold nanorods were synthesised as evidenced by the UV-Vis spectrum recorded, see Fig. 4e.

The two-peak spectrum of these rods shows a longitudinal peak at ~760 nm and a transverse peak at ~525 nm. Once again, all reagents, five in this case, were free to be varied by the algorithm. The formula used to calculate the fitness factor in this case used the relative areas of specific nm regions under the spectra curve to determine the fitness and is shown in Fig. 4d. The system first searched for these two peaks and then aimed to increase the peak intensity of the longitudinal peaks relative to the transverse, whilst also minimising the absorbance in the region between these peaks (Fig. 4e). If this value of the transverse peak was more than x0.75 intense as the absorbance of the longitudinal peak, the system assigned a fitness value of 0 to this experiment. The reaction temperature for the nanorod synthesis in the automated system was set at 30 °C and each vial was stirred for 30 min and left undisturbed for additional 60 min before UV-vis sample extraction. The experimental platform for nanorods optimisation ran for 10 generations with 15 experiments per generation.

The fitness progression shown in Fig. 4d shows an upward trend in fitness factor over these 10 generations, and this indicates the robot platform had produced results similar to those seen on the bench using literature values (780 nm longitudinal peak). Table 1 compares the difference in reagent addition from the literature values to those discovered by the platform (stock volumes normalised to those used in the literature). An aspect of this space that should be highlighted is that the error in the fitness values can be larger for this synthesis than those seen for spheres. A reasonable explanation for this is the inherent difficulties in nanoparticle synthesis in that small changes in the recipe for the formation of gold nanorods can lead to significant differences in the reaction products. Despite this, the system clearly obtained an upward trend towards the target. The platform learned to synthesise AuNRs by using slightly higher volumes of $AgNO_3$ and ascorbic acid, similar quantities of seeds and CTAB, and significantly less $HAuCl_4$.

**Table 1 Comparison of reagents used in optimal synthesis for space 2 (nanorod target) with the platform vs synthesis described in the literature (normalised to literature stock solutions).**

| Reagent | Manual V (mL) | Normalised robot V (mL) | Difference (mL) |
|---|---|---|---|
| $HAuCl_4$ | 5 | 2.54 | 2.46 |
| CTAB | 5 | 4.45 | 0.55 |
| $AgNO_3$ | 0.15 | 0.23 | 0.08 |
| Ascorbic A | 0.07 | 0.095 | 0.025 |

The UV-Vis spectra shown in Fig. 4e reveals a good similarity between literature and the optimised synthesis protocols produced using the evolutionary algorithm. This shows that the system can proceed efficiently towards a pre-defined objective using a simple mathematical formulation for comparing two spectra. The highest scoring sample was centrifuged at 12,000 RPM for 10 min and washed with ultra-pure water twice. TEM of the sample shown Fig. 4f corresponds to the resulting precipitate from solution that gave the UV-vis spectrum seen in Fig. 4e i.e. the highest optimised rods obtained by the platform. The image shows several rods with sizes between 45 and 55 nm, and the average aspect ratio from 140 nanoparticles in this sample was 3.97 ± 0.73. UV of samples in early generations of this space indicated the presence of cubic particles growing alongside some rods, this formation decreased progressively throughout the generation series (See Supplementary information, Au Nanorods).

**Single peak target**. In the next stage of our investigations we decided to explore an unknown shape regime whereby we no longer used literature values for a desired and known shape outcome. To achieve this, we chose a new objective whereby the desired UV-Vis spectra was set to have a maximum peak at 580 nm. We chose this objective based on the peak position, discarding any other features of the spectra in the hope of producing an unknown outcome. For this run we continued to run platform generations of 15 reactions until the fitness level remained constant. Eight generations in total were required. The stock reagents that were used for this set of experiments were the same as for the synthesis of AuNRs, with the exception of the ≈2 nm spherical seeds. The seeds for this series were the rods produced from space 2. In total, 24 identical reaction using the optimised reaction conditions from space 2 were performed on the platform, the particles cleaned and dispersed the same volume as the original reactions (240 mL total). These were aqueous solutions of CTAB (0.2 M), $HAuCl_4$ (0.001 M), $AgNO_3$ (0.00005 M) and ascorbic acid (0.0065 M). The reaction conditions and protocols were kept identical to the previous runs with the algorithm free to change all of the reagents excluding the seed solution that was fixed to 1 mL per reaction due to its relative difficulty to prepare. In the similar fashion as before Fig. 4g–i show the progression of this final stage; from the progress made by the GA, the observed vs objective peak position and the TEM image of the resulting octahedral shaped particles. The highest fitness sample was produced from a solution of 0.479 mL CTAB, 1.813 mL $HAuCl_4$, 3.327 mL $AgNO_3$, 3.38 mL ascorbic acid and 1 mL of seeds All products can be readily reproduced using the automated platform and the manual synthesis of these products was carried on the bench, enacting the precise formula discovered by the robot, out in order to determine if the results obtained by the

automated system were reproducible by a chemist, and all the products were reproduced successfully.

## Discussion

Although the synthesis of AuNPs of different size and shape has been studied before, our work presents a new methodology to further and advance this field by using the unbiased nature of algorithmically driven synthesis in a closed loop robot platform. The platform presented in this paper has been able to synthesise complex nanomaterials starting from simple, raw chemicals by a process of hierarchical evolution. Our system has demonstrated for the first time, seed mediated nanoparticle synthesis assisted by an evolutionary algorithm in a controlled and reproducible manner. This automated, closed loop approach allows us not only to create known architectures reliably but also could be used as a tool to discover complex nano-constructs using desired spectroscopic responses. Lower tier nanoparticles were fed into the system in order to obtain more complex structures. This methodology, whilst offering the benefits of automation; speed, safety, reproducibility, etc. provides the chemist with a tool for developing new synthetic methods and the potential for new discoveries. These discoveries could lead to a better understanding of how nanoparticles are formed and to develop new application areas by searching for a given property, as well as ensuring that complex faceted nanoparticles can be reproduced easily using a digital code in an automatic platform[32,33].

## Methods

Methods including statements of data availability and any associated accession codes and references, are available in the online version of this paper.

## Data availability

Due to the quantity produced during this work, the full raw data from the robot is available only on request.

## Code availability

The source code is available at https://github.com/croningp/NanomaterialsDiscovery.

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

## Acknowledgements

We gratefully acknowledge financial support from the EPSRC (Grant Nos EP/H024107/1, EP/I033459/1, EP/J00135X/1, EP/J015156/1, EP/K021966/1, EP/K023004/1, EP/K038885/1, EP/L015668/1, EP/L023652/1), and the ERC (project 670467 SMART-POM). We gratefully acknowledge Margaret Mullin who performed all TEM analysis, preparing samples and running the microscope

## Author contributions

L.C. devised the concept and the initial algorithm and platform. The platform was built by D.S., G.K. and J.G.; D.S. and G.K. helped collect data. S.M. helped in the early stages of the project. J.G. and A.S. helped L.C. coordinate the team. All the co-authors helped write the manuscript with L.C.

## Competing interests

The authors declare no competing interests.
