## [Peer Review File · Nature Communications]

Reviewers' comments:

Reviewer #1 (Remarks to the Author):

The authors report an autonomous platform to explore the chemical space of gold nanoparticles. They built a robotic system to prepare and analyse optically the performance of the particle synthesis, which is suggested by a genetic algorithm. The whole workflow is controlled by an in-house developed software written in python/C++.

Overall the manuscript is enjoyable to read and presents an innovative approach to the discovery process. I would recommend publication after addressing the following suggestions and comments:

1. The size of Space 2 and 3 is unclear to me. From Figure 1, it looks like there are four parameters to play with. However, later in the text - "after gentle mixing, ascorbic acid (70 μ l, 0.0788 M) was added" - and based on Table 2, it looks like Space 2 and 3 have three free parameters to play with. Could you please confirm and clarify?
2. There is a total of 100 (10x10) experiments in Space 1, 150 (10x15) experiments in Space 2, and 150 (10x15) experiments in Space 3. That sounds like a lot of experiment in order to have a good fitness, i.e., minimise the loss function. In your opinion, how would that compare with other approaches to designing experiment? Does that include the initial set of randomly generated parameters? Could you comment and explain why Space 1 is explored in batches of 10, while Space 2 & 3 are explored in batches of 15?
3. In the absorbance spectra of Space 2, how did you decide on the threshold factor of 0.75 between the median peak and the target peak to assign a non-zero fitness value?
4. For Space 3, how do you make sure that all 150 experiments yield arrow-headed nanorods and not nanosphere? The absorption peak of the latter (572nm) seems very close to the one of the nanospheres at 573.61nm, and to me, spectrum S7/S8/S10/S11/S13/S14 (nanosphere) are very similar to S25 (expanded search).
5. Minor comment: introduce the acronym AuNPs before using it (bottom of p.2)
6. Minor comment: In Table 1, the "Difference (mL)" for "CTAB" should read 0.56

Reviewer #2 (Remarks to the Author):

The authors describe a self-optimizing reactor for the production of gold nanoparticles of varying shape. The paper describes an interesting application of self-optimizing reactors, but is weakened by a significant lack of experimental/procedural detail as set out below.

There is a sizable body of papers on self-optimizing reactors, and it is surprising that none of these has been cited in the manuscript. Key papers by the groups of Bourne, de Mello, Jensen, Poliakoff and Rueping for instance are missing. The use of genetic algorithms (GAs) distinguishes the current manuscript from previous papers, but the justification for their use is not well made. The authors comment on the suitability of GAs "for searching through large and complex data sets... when a large number of variables must be explored", but neither of these considerations applies to the reported work, which involves just "10 generations with 10 experiments per generation". There are many possible algorithms that could be sensibly employed for an optimization of the size described. If there is a genuine advantage to using a genetic algorithm for the current application, this should be described in detail with comparative experimental studies against other optimization procedures.

In general the paper lacks sufficient experimental detail for the reader to understand exactly what has been done or reproduce the work. For instance, it is unclear what specific reaction parameters were used to control the optimization. What were the precise compositions of the stock solutions used? How were the automated syntheses carried out? Was it purely the volumes of the stock solutions that were altered or were the concentrations also changed? It is very difficult to infer the experimental procedure from the limited information given.

The specific algorithm used for the optimization was not stated. The only description I could find was a statement in the supporting information, saying "The algorithm used within this system was a standard optimization genetic algorithm – a form of evolutionary algorithm", which is an inadequate description. The exact algorithm used should be stated, together with all relevant optimization parameters. The paper is unclear on the number of experiments carried out for each generation. In some places, it says 15 and in others it says 10.

The authors state "The final UV-vis spectra shown in Figure 4d shows an upward trend in fitness factor over these 10 generations, and this indicates the robot platform had produced results all but identical to those seen on the bench using literature values." I assume the authors mean 4e, here? If so, then it is inaccurate to describe the results as "all but identical". The peak positions and ratio of peak heights differ substantially for the two spectra.

The authors conclude by saying "Our system has demonstrated for the first time, seed mediated nanoparticle synthesis assisted by an evolutionary algorithm in a controlled and reproducible manner" but no data is provided on the reproducibility of the system to support this statement.

The labeling of the raw spectra in the SI is uninformative. For instance, the description "Seed 128, Generation 0000, Experiment 0003" provides little information of value. What specific reaction conditions were used to obtain each spectrum?

Overall, the manuscript lacks the experimental detail required to support the claims made. Therefore I do not recommend the paper for publication in Nature Communications, and suggest it is submitted

elsewhere once the above issues have been addressed.

Reviewer #3 (Remarks to the Author):

In this manuscript, the authors demonstrate a semi-batch liquid handling robotic platform for optimize the synthetic conditions of Au nanomaterials of different morphologies. Au spherical nanoparticles, rods, and anisotropic arrow-faceted nanoparticles were chosen as the demonstrated targets of the robot. While the Au nanoparticles were used as the seeds for the preparation of Au nanorods, the rods were used as the seeds to synthesis the arrow-faceted nanoparticles. UV-Vis spectra were measured and used as feedback signals to evaluate how the products' shapes were close to the desired target. Overall, the work should be interesting for publication in Nat. Commun. I would like to recommend its publication after the following issues are addressed:

1. More TEM images should be provided to show how the quality of three different nanomaterials was getting improved with increased generation. As for the Au nanorods and arrows, the TEM images should be of lower magnifications to display large-area samples instead of just several particles in the images.
2. As for the sample of Au nanorods, both the UV-Vis (absorption at $\sim 510\text{nm}$) and TEM data indicated that there should be a large amount of spherical nanoparticles. So the authors need to make comments on this. The information on the ratio of different-shaped particles should be given.
3. The robot optimized the protocols for the three different-shaped nanomaterials whose syntheses have been already optimized by batch reactions. I would encourage the authors to optimize conditions for some structures whose syntheses have not been optimized. Typically, in every batch of synthesizing anisotropic nanoparticles, there are always some minor products with weird morphologies. It would be very interesting to see the robot can be used for the optimization of those minor products.

Reviewers' comments in italics, our responses in normal font.

Reviewer #1:

The authors report an autonomous platform to explore the chemical space of gold nanoparticles. They built a robotic system to prepare and analyse optically the performance of the particle synthesis, which is suggested by a genetic algorithm. The whole workflow is controlled by an in-house developed software written in python/C++.

Overall the manuscript is enjoyable to read and presents an innovative approach to the discovery process. I would recommend publication after addressing the following suggestions and comments:

1. The size of Space 2 and 3 is unclear to me. From Figure 1, it looks like there are four parameters to play with. However, later in the text - "after gentle mixing, ascorbic acid (70µl, 0.0788 M) was added" - and based on Table 2, it looks like Space 2 and 3 have three free parameters to play with. Could you please confirm and clarify?

The initial space used for nanosphere optimisation had four parameters to choose from (HAuCl₄, Ascorbic acid, seeds, and CTAB). Space 2 and 3 had five parameters (HAuCl₄ CTAB AgNO₃ Ascorbic Acid, and Seed volume). This has been clarified and added to the text.

2. There is a total of 100 (10x10) experiments in Space 1, 150 (10x15) experiments in Space 2, and 150 (10x15) experiments in Space 3. That sounds like a lot of experiment in order to have a good fitness, i.e., minimise the loss function. In your opinion, how would that compare with other approaches to designing experiment? Does that include the initial set of randomly generated parameters? Could you comment and explain why Space 1 is explored in batches of 10, while Space 2 & 3 are explored in batches of 15?

The target of space one is a single UV target and a lower complexity shape and therefore required less trial and error on the machines parts to reach. It was believed 8 x 15 was sufficient for this optimization. Yes this total includes the random set in the initial generation. Given the size of the spaces, space 1 being four dimensional and 2 +3 being five we believe this reaction numbers are quite reasonable. The platform was free to choose any volume of these reagents and considering the volume resolution of our tricontinent syringe pumps, the spaces are enormous. Therefore, we believe 120-150 reactions to achieve our targets with no priors in the algorithm is reasonable.

3. In the absorbance spectra of Space 2, how did you decide on the threshold factor of 0.75 between the median peak and the target peak to assign a non-zero fitness value?

The threshold value of 0.75 was chosen to minimise the influence of peaks that resided between the lower and upper regions (572nm and 850nm respectively). The actual value of 75% was an arbitrary value that seemed to be sufficient. Lower would most likely be too aggressive and any higher would allow sub-optimal spectra to creep through.

4. For Space 3, how do you make sure that all 150 experiments yield arrow-headed nanorods and not nanosphere? The absorption peak of the latter (572nm) seems very close to the one of the nanospheres at 573.61nm, and to me, spectrum S7/S8/S10/S11/S13/S14 (nanosphere) are very similar to S25 (expanded search).

This was always a concern however, it was believed that the chances of reverting back to a spherical shape having used rods as the seeds for the experiment was highly unlikely/impossible) and the TEM revealed the presence of arrowheads rods. Spherical particles could exist in the sample, having formed

independently however given the weak reducing agent and presence of rods seeds we believe it highly unlikely.

5. *Minor comment: introduce the acronym AuNPs before using it (bottom of p.2)*

This has been modified.

6. *Minor comment: In Table 1, the "Difference (mL)" for "CTAB" should read 0.56*

Due to the modifications of the paper this table has now been removed as it is no longer relevant to the chemistry of space one.

Reviewer #2:

The authors describe a self-optimizing reactor for the production of gold nanoparticles of varying shape. The paper describes an interesting application of self-optimizing reactors but is weakened by a significant lack of experimental/procedural detail as set out below.

1. There is a sizable body of papers on self-optimizing reactors, and it is surprising that none of these has been cited in the manuscript. Key papers by the groups of Bourne, de Mello, Jensen, Poliakoff and Rueping for instance are missing. The use of genetic algorithms (GAs) distinguishes the current manuscript from previous papers, but the justification for their use is not well made. The authors comment on the suitability of GAs "for searching through large and complex data sets... when a large number of variables must be explored", but neither of these considerations applies to the reported work, which involves just "10 generations with 10 experiments per generation". There are many possible algorithms that could be sensibly employed for an optimization of the size described. If there is a genuine advantage to using a genetic algorithm for the current application, this should be described in detail with comparative experimental studies against other optimization procedures.

We have introduced the area better and have added an introductory reference from Bourne to optimisation, then on to catalysis by Rueping, and then the crystal nucleation work of deMello. On the second point, the number of reactions is not what was referred to as the 'large number of variables', it was the fact that the reagent quantities making up the space could all be varied by the algorithm making the space of **potential** reaction enormous, each reactant adding another dimension to the space search.

2. In general the paper lacks sufficient experimental detail for the reader to understand exactly what has been done or reproduce the work. For instance, [1] it is unclear what specific reaction parameters were used to control the optimization. [2] What were the precise compositions of the stock solutions used? [3] How were the automated syntheses carried out? [4] Was it purely the volumes of the stock solutions that were altered or were the concentrations also changed? [5] It is very difficult to infer the experimental procedure from the limited information given. [note we added the numbers to reference below]

[1] The target UV signal was used to control the optimization. [2] The stock solutions for spherical particles now listed clearly in the result section and SI and for rods they are listed clearly (page 9). [3] We feel figure 2 and 3 detail the process is sufficient detail. [4] There is no mention of stock concentration change and we believe it is clear that yes volume changes were the only change allowed by the platform. [5] With the addition of the flow diagram of the exact automated procedure we believe this will clear the issue.

The specific algorithm used for the optimization was not stated. The only description I could find was a statement in the supporting information, saying "The algorithm used within this system was a

standard optimization genetic algorithm – a form of evolutionary algorithm”, which is an inadequate description. The exact algorithm used should be stated, together with all relevant optimization parameters. The paper is unclear on the number of experiments carried out for each generation. In some places, it says 15 and in others it says 10.

The algorithm is detailed in full in the SI. The algorithm itself employs the six stages detailed in the SI and does not have a specific name other than Genetic algorithm – it is a generalised term to describe the algorithm. Optimisation parameters used are the quantities of reagents used during each experiment. This can be made clearer in the manuscript. The number of experiments was increased due to a fundamental change in the platform’s physical structure. The difference between 10 and 15 experiments per generation, whilst noticeable would not have a major impact on the end results, more experiments would have been conducted with similar optimisation values.

The authors state “The final UV-vis spectra shown in Figure 4d shows an upward trend in fitness factor over these 10 generations, and this indicates the robot platform had produced results all but identical to those seen on the bench using literature values.” I assume the authors mean 4e, here? If so, then it is inaccurate to describe the results as “all but identical”. The peak positions and ratio of peak heights differ substantially for the two spectra.

The phrasing has been amended from ‘all but identical’ to ‘similar to’. Figure 4d shows significant fitness increase across the reaction generations, and therefore ‘indicates’ progress toward particles similar to those found in the literature.

The authors conclude by saying “Our system has demonstrated for the first time, seed mediated nanoparticle synthesis assisted by an evolutionary algorithm in a controlled and reproducible manner” but no data is provided on the reproducibility of the system to support this statement.

All of our optimized reactions have been reproduced both on the platform and the bench (keeping in mind this is how we obtain samples for TEM as the samples are removed to waste during the platforms operation). The software, platform design, and parameters used to repeat this system can be made available upon request. Also a Git repository will be made available for extensive details of the platform construction and code.

The labeling of the raw spectra in the SI is uninformative. For instance, the description “Seed 128, Generation 0000, Experiment 0003” provides little information of value. What specific reaction conditions were used to obtain each spectrum?

This formatting has been removed from the SI, leaving more relevant information in the SI

Overall, the manuscript lacks the experimental detail required to support the claims made. Therefore, I do not recommend the paper for publication in Nature Communications, and suggest it is submitted elsewhere once the above issues have been addressed.

We can only disagree with this statement and given our responses above, think our disagreement justified. The manuscript clearly shows an overall trend towards a different desired targets over the generations conducted using the automated platform.

Reviewer #3:

In this manuscript, the authors demonstrate a semi-batch liquid handling robotic platform for optimize the synthetic conditions of Au nanomaterials of different morphologies. Au spherical nanoparticles, rods, and anisotropic arrow-faceted nanoparticles were chosen as the demonstrated targets of the robot. While the Au nanoparticles were used as the seeds for the preparation of Au nanorods, the rods

were used as the seeds to synthesis the arrow-faceted nanoparticles. UV-Vis spectra were measured and used as feedback signals to evaluate how the products' shapes were close to the desired target. Overall, the work should be interesting for publication in Nat. Commun. I would like to recommend its publication after the following issues are addressed:

1. More TEM images should be provided to show how the quality of three different nanomaterials was getting improved with increased generation. As for the Au nanorods and arrows, the TEM images should be of lower magnifications to display large-area samples instead of just several particles in the images.

The authors believe that given the level of variability in the system through a given series of generations is very high, we don't we expect a linear product formation for individual experiments, we expect average increase in progress across full generations. For spheres, a simpler particle, this variability is not as pronounced due to the nature of isotropic growth in the reactions. As such progressive TEMs of the spherical space have been added to the SI. These show progression from particle mixtures containing spheres to highly pure but smaller spherical particles to our final desired product spheres. In more complex systems such as rods we have not recorded intermediate generations as TEM only became relevant when the spectral target was obtained. More TEMs have been added to the SI showing larger populations of nanoparticles, see page S23-26.

2. As for the sample of Au nanorods, both the UV-Vis (absorption at ~510nm) and TEM data indicated that there should be a large amount of spherical nanoparticles. So the authors need to make comments on this. The information on the ratio of different-shaped particles should be given.

There will inevitably be a quantity of spherical nanoparticles present as a sufficient quantity of nanorods present as well. When exploring without prior knowledge on the part of the algorithm it is expected that not all seeds will go to completion towards nanorods. This will be added to the manuscript.

3. The robot optimized the protocols for the three different-shaped nanomaterials whose syntheses have been already optimized by batch reactions. I would encourage the authors to optimize conditions for some structures whose syntheses have not been optimized. Typically, in every batch of synthesizing anisotropic nanoparticles, there are always some minor products with weird morphologies. It would be very interesting to see the robot can be used for the optimization of those minor products.

Thank you for the suggestion. We are continuing this work on an improved platform and therefore this will be addressed in future work.

Reviewers' comments:

Reviewer #1 (Remarks to the Author):

The authors report an autonomous platform to explore the chemical space of gold nanoparticles. They built a GA-driven robotic system to prepare and analyze optically the performance of the particle synthesis. The workflow is controlled by an in-house developed software.

The authors addressed all my questions and concerns, and the paper is ready to be published.

Reviewer #2 (Remarks to the Author):

The authors have made a number of changes to improve the readability of the paper. However, the concerns raised in my previous review remain. The introduction still inadequately cites the existing literature on self-optimizing reactors. The justification for using a genetic algorithm remains weak. The authors have added missing information relating to the experimental procedure. However, the description of the genetic algorithm is still incomplete, lacking for instance information concerning the cross-over probabilities, mutation probabilities and penalty functions. As noted, in my previous comments, Fig. 4d does not show “a final UV/Vis spectrum” – presumably the authors mean Fig. 4e? As far as I can tell, the manuscript still contains no data to support the claims made about reproducibility. The labelling of the raw spectra in the SI is unchanged and remains uninformative – what were the experimental conditions corresponding to these spectra? The claim in the abstract of “discovering entirely new systems” seems to me to be overstated.

Reply to reviewer #2 Concerning “A Nanomaterials Discovery Robot for the Darwinian Evolution of Shape Programmable Gold Nanoparticles”

We thank the reviewer for their additional questions which we have answered without exception and think their questions have helped us improve the manuscript.

Comment 1: The authors have made a number of changes to improve the readability of the paper. However, the concerns raised in my previous review remain. The introduction still inadequately cites the existing literature on self-optimizing reactors.

Response 1: To address this we have added another set of references, 26-28, chosen to highlight the use of algorithms for self-optimising of chemistry.

Comment 2: The justification for using a genetic algorithm remains weak. The authors have added missing information relating to the experimental procedure. However, the description of the genetic algorithm is still incomplete, lacking for instance information concerning the cross-over probabilities, mutation probabilities and penalty functions.

Response 2: The SI now contains all information about how the GA progresses from one generation to the next. This new section details the selection method (including the means we use to ensure the same reaction is not selected across multiple generations), the crossover and mutation probabilities, penalties and fitness functions used for each reaction space.

Comment 3: As noted, in my previous comments, Fig. 4d does not show “a final UV/Vis spectrum” – presumably the authors mean Fig. 4e?

Response 3: This have been corrected

Comment 4: As far as I can tell, the manuscript still contains no data to support the claims made about reproducibility. The labelling of the raw spectra in the SI is unchanged and remains uninformative – what were the experimental conditions corresponding to these spectra?

Response 4: To address this we have added new TEM data, gathered using reproduced synthesis as the original samples during the GA driven reaction series are disposed of by the automated platform. To create the seeds for space 3, the optimised conditions for rods are used. This involves performing 24 identical reactions using the platform to produce enough seeds for the complete reaction generation series of space 3. Each of these 24 reactions would produce rods with a longitudinal peak at 780 nm, when the 24 reactions were combined the full solution yielded a peak at 777 nm. This allows us to check and prove reproducibility of the systems produced by the platform. This spectrum can be seen in the SI. With regards to the spectra in the SI used to show the progression toward the goal, the generation they occurred in the figure captions and the contents of the reaction solutions are now included in individual tables for each space.

Comment 5: The claim in the abstract of “discovering entirely new systems” seems to me to be overstated.

We have modified the abstract to now read

“Here we present an autonomously driven materials-evolution robotic platform that allows us to reliably discover the conditions to produce gold-nanoparticles that can run for many cycles, discovering entirely new synthetic conditions for known nanoparticle shapes using the opto-electronic properties as a driver”